# TREE-STRUCTURED VARIATIONAL AUTOENCODER

**Richard Shin**[*]
Department of Electrical Engineering and Computer Science
University of California, Berkeley
`ricshin@cs.berkeley.edu`

**Alexander A. Alemi, Geoffrey Irving & Oriol Vinyals**
Google Research, Google Brain, Google DeepMind
`{alemi,geoffreyi,vinyals}@google.com`

## ABSTRACT

Many kinds of variable-sized data we would like to model contain an internal hierarchical structure in the form of a tree, including source code, formal logical statements, and natural language sentences with parse trees. For such data it is natural to consider a model with matching computational structure. In this work, we introduce a variational autoencoder-based generative model for tree-structured data. We evaluate our model on a synthetic dataset, and a dataset with applications to automated theorem proving. By learning a latent representation over trees, our model can achieve similar test log likelihood to a standard autoregressive decoder, but with the number of sequentially dependent computations proportional to the depth of the tree instead of the number of nodes in the tree.

## 1 INTRODUCTION

A significant amount of recent and ongoing work has explored the use of neural networks for modeling and generating various kinds of data. Newer techniques like the variational autoencoder (Rezende et al., 2014; Kingma & Welling, 2013) and generative-adversarial networks (Goodfellow et al., 2014) enable training of graphical models where the likelihood function is a complicated neural network which normally makes it infeasible to specify and optimize the marginal distribution analytically. Another family of techniques involves choosing an ordering of the dimensions of the data (which is particularly natural for sequences such as sentences) and training a neural network to estimate the distribution over the value of the next dimension given all the dimensions we have observed so far.

These techniques have led to significant advances in modeling images, text, sounds, and other kinds of complicated data. Language modeling with sequential neural models have halved perplexity (roughly, the error at predicting each word) compared to n-gram methods (Jozefowicz et al., 2016). Neural machine translation using sequence-to-sequence methods have closed half of the gap in quality between prior machine translation efforts and human translation (Wu et al., 2016). Generative image models have similarly progressed such that they can generate samples largely indistinguishable from the original data, at least for relatively small and simple images (Gregor et al., 2015; 2016; Kingma et al., 2016; Salimans et al., 2016; van den Oord et al., 2016), although the quality of the model here is harder to measure in an automated way (Theis et al., 2015).

However, many kinds of data we might wish to model are naturally structured as a tree. Computer program code follows a rigorous grammar, and the usual first step in processing it involves parsing it into an abstract syntax tree, which simultaneously discards aspects of the code irrelevant to the semantics such as whitespace and extraneous parentheses, and makes it more convenient to further interpret, analyze, or manipulate. Statements made in formal logic similarly have a hierarchical structure, which determines arguments to predicates and functions, the scoping of variables and quantifiers, and the application of logical connectives. Natural language sentences also contain a latent syntactic structure which is necessary for determining the meaning of the sentence, although

---

[*]Majority of work done while at Google.

most sentences admit many possible parses due to the multiple meanings of each word and ambiguous groupings.

In this paper, we explore how we can adapt the variational autoencoder to modeling tree-structured data. In general, it is possible to treat the tree as a sequence and then use a sequential model. However, a model which follows the structure of the tree may better capture long-range dependencies: recurrent models sometimes have trouble learning to remember and use information from the distant past when it is relevant to the current context, but these distant parts of the input may be close to each other within the tree structure.

In our proposed approach, we decide the identity of each node in the tree by using a top-down recursive neural network, causing the distributed representation which decides the identity of each node in the tree to be computed as a function of the identity and relative location of its parent nodes.

By using the architecture of the variational autoencoder (Rezende et al., 2014; Kingma & Welling, 2013), our model can learn to capture various features of the trees within continuous latent variables, which are added as further inputs into the top-down recursive neural network and conditions the overall generation process. These latent variables allow us to generate different parts of the tree in parallel; specifically, given a parent node $n$ and its children $c_1$ and $c_2$, the generation of (the distribution placed over different values of) $c_1$ and its descendants is independent of the generation of $c_2$ and its descendants (and vice versa), once we condition upon the latent variables. By structuring the model this way, while our model generates one dimension (the identity of each node within the tree) of the data a time, it is not autoregressive as the probability distribution for one dimension is not a function of the previously generated nodes.

We evaluate our model on a variety of datasets, some synthetic and some real. Our experimental results show that it achieves comparable test set log likelihood to autoregressive sequential models which do not use any latent variables, while offering the following properties:

- For balanced trees, generation requires $O(\log n)$ rather than $O(n)$ timesteps required for a sequential model because the children of each node can be generated in parallel.
- It is straightforward to resample a subtree while keeping the other parts of the tree intact.
- The generated trees are syntactically valid by construction.
- The model produces a latent representation for each tree, which may prove useful in other applications.

## 2 BACKGROUND AND RELATED WORK

### 2.1 TREE-STRUCTURED MODELS

Recursive neural nets, which processes a tree in a bottom-up way, have been popular in natural language processing for a variety of tasks, such as sentiment analysis (Socher et al., 2013), question answering (Iyyer et al., 2014), and semantic relation extraction (Socher et al., 2012). Starting from the leaves of the tree, the model computes a representation for each node by combining the representations of its child nodes. In the case of natural language processing, each tree typically represents one sentence, with the leaf nodes corresponding to the words in the sentence and the structure of the internal nodes determined by the constituency parse tree for the sentence.

If we restrict ourselves to binary trees (given that it is possible to binarize arbitrary trees in a lossless way), the we compute the $k$-dimensional representation $\mathbf{r}_n \in \mathbb{R}^k$ for a node $n$ by combining the representations of nodes $n_{\text{left}}$ and $n_{\text{right}}$:

$$r_n = f(W\mathbf{r}_{n_{\text{left}}} + V\mathbf{r}_{n_{\text{right}}})$$

where $W$ and $V$ are square matrices (in $\mathbb{R}^{d \times d}$) and $f$ is a nonlinear activation function, applied elementwise. Leaf nodes are represented by embedding the content of the leaf node into a $d$-dimensional vector, by specifying a lookup table from words to embedding vectors for instance.

Variations and extensions of this approach specify more complicated relationships between $\mathbf{r}_n$ and the childen's representations $\mathbf{r}_{n_{\text{left}}}$ and $\mathbf{r}_{n_{\text{right}}}$, or allow internal nodes to have variable numbers of children. For example, Tai et al. (2015) extend LSTMs to tree-structured models by dividing the vector

representation of each node into a memory cell and a hidden state and updating them accordingly with gating.

Neural network models which generate or explore a tree top-down have received less attention, but have been applied to generation and parsing tasks. Zhang et al. (2015) generate natural language sentences along with their dependency parses simultaneously. In their specification of a dependency parse tree, each word is connected to one parent word and has a variable number of left and right children (a depth-first in-order traversal on this tree recovers the original sentence). Their model generates these trees by conditioning the probability of generating each word on its ancestor words and the previously-generated sibling words using various LSTMs. Dyer et al. (2016) generate sentences jointly with a corresponding constituency parse tree. They use a shift-reduce parser architecture where *shift* is replaced with word-generation action, and so the sentence and the tree can be generated by performing a sequence of actions corresponding to a depth-first pre-order traversal of the tree. Each action in the sequence is predicted based upon the tree constructed so far, the words (the tree's terminal nodes) generated so far, and the previous actions performed. Dong & Lapata (2016) generate tree-structured logical forms using LSTMs, where the LSTM state branches along with the tree's structure; they focus on generating these logical forms when conditioned upon a natural-language description of it.

## 2.2 VARIATIONAL AUTOENCODERS

The variational autoencoder (Kingma & Welling, 2013; Rezende et al., 2014), or VAE for short, provides a way to train a generative model with a fixed prior $p(\mathbf{z})$ and a neural network used to specify $p_\theta(\mathbf{x} \mid \mathbf{z})$. Typically, the prior $p(\mathbf{z})$ is taken to be a standard multivariate normal distribution (mean at 0) with diagonal unit covariance. Naively, in order to optimize $\log p(\mathbf{x})$, we need to compute the following integral:

$$\log p_\theta(\mathbf{x}) = \log \int_{\mathbf{z}} p_\theta(\mathbf{x} \mid \mathbf{z}) p(\mathbf{z}) d\mathbf{z}$$

which can be tractable when $p_\theta(\mathbf{x} \mid \mathbf{z})$ is simple but not when we want to use a neural network to represent it. Inference of the posterior $p(\mathbf{z} \mid \mathbf{x})$ also becomes intractable.

Instead, we learn a second neural network $q_\phi(\mathbf{z} \mid \mathbf{x})$ to approximate the true posterior, and use the following variational bound:

$$\log p(\mathbf{x}) \geq -D_{KL}(q_\phi(\mathbf{z} \mid \mathbf{x}) \,\|\, p(\mathbf{z})) + \mathbb{E}_{q_\phi(\mathbf{z}|\mathbf{x})}[\log p_\theta(\mathbf{x} \mid \mathbf{z})]$$

where $D_{KL}$ represents the Kullback-Leibler divergence between the two distributions. Given that we represent $q_\phi(z \mid x)$ with a neural network which outputs the mean and diagonal covariance for a normal distribution, we can analytically compute the KL divergence term and then use the reparameterization trick:

$$\mathbb{E}_{q_\phi(\mathbf{z}|\mathbf{x})}[\log p_\theta(\mathbf{x} \mid \mathbf{z})] = \mathbb{E}_{p(\epsilon)}[\log p_\theta(\mathbf{x} \mid \mathbf{z} = \mu + \sigma \cdot \epsilon)]$$

where $p(\epsilon)$ is a standard multivariate normal distribution, and $\mu$ and $\sigma$ are outputs of the neural network implementing $q_\phi(\mathbf{z} \mid \mathbf{x})$.

These two techniques combined allow us to compute stochastic gradients (by sampling $\epsilon$, treating it as constant, and backpropagating through the model) and use standard neural network training techniques (such as SGD, Adagrad, and Adam) to train the model.

Another interpretation of the variational autoencoder follows from a modification of the regular autoencoder, where we would like to learn a mapping $x \rightarrow z$ from the data to a more compact representation $z$, and an inverse mapping $z \rightarrow x$. In the VAE, we replace the deterministic $x \rightarrow z$ with a probabilistic $q(z \mid x)$, and as a form of regularization, we ensure that this distribution is close to a prior $p(z)$.

## 3 TREE-STRUCTURED VARIATIONAL AUTOENCODER

In this section, we describe how we combine the variational autoencoder and recursive neural networks in order to build our model.

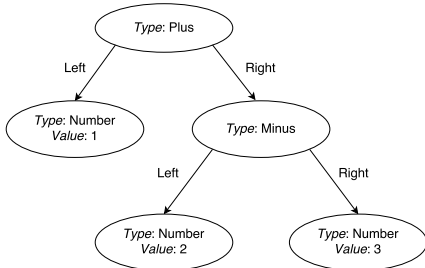

Figure 1: An example tree for $1 + 2 - 3$. The binary operators are represented with non-terminal nodes, with two required children "left" and "right". The numbers are terminal nodes.

### 3.1 TREES

We consider arbitrarily-branching typed trees where each node contains a *type*, and either *child nodes* or a terminal *value*. Each type may be a *terminal* type or a *non-terminal* type; nodes of terminal type contain a value, and nodes of non-terminal type have (zero or more) child nodes.

A non-terminal type $T$ comes with a specification for how many children a node $N_T$ of type $T$ should have, and the types permissible for each child location. We distinguish three types of child nodes:

- $N_T$ may have some number of *singular* child nodes. For the $i$th singular child, we specify $\text{SINGULARCHILD}(T, i) = \{T_1, \cdots, T_n\}$ as the set of types that child node can have. If the singular child node is optional, we denote this by including $\phi$ in this set. $\text{SINGULARCHILDCOUNT}(T)$ gives the number of singular child nodes in $T$.

- $N_T$ may have an arbitrary number of *repeated* child nodes. Each repeated child node must have type belonging within $\text{REPEATEDCHILDREN}(T) = \{T_1, \cdots\}$. If this set is empty, no repeated child nodes are allowed. These children may be of different types.

For each terminal type, we have a list of values that a node of this type can have. We also have a list of types that the root node can have.

The above specification serves as an extension of *context-free grammars*, which are commonly used to specify formal languages. The main difference is in optional and repeated children, which makes it easier to specify an equivalent grammar with fewer non-terminal types.

As an example, consider the `for` loop in the C programming language. A node representing this contains three singular children: an initializer expression, the condition expression (evaluated to check whether the loop should continue running), and the iteration statement, which runs at the end of each loop iteration. It also has repeated children, one child per statement in the loop body.

### 3.2 BUILDING A TREE

Now that we have specified the kinds of trees we consider, let us look at how we might build one. First, we describe the basic building block that we use to create one node, then we will look at how to compose these together to build an entire tree.

Assume that we know the node that we are about to construct should have type $T$, and that we have a hidden state $\mathbf{h} \in \mathbb{R}^k$ which contains further information about the node.

- If $T$ is a terminal type, we use $\text{WHICHTERMINALVALUE}_T(h)$, producing a probability distribution over the set of possible values, and sample from this distribution to choose the value.

- If $T$ is a non-terminal type, we use the following procedure $\text{GENERATENODE}(T, \mathbf{h})$:
    1. Compute $m = \text{SINGULARCHILDCOUNT}(T) + \mathbb{1}\{\text{REPEATEDCHILDREN}(T) \neq \emptyset\}$. In other words, count the number of singular children, and add 1 if the type allows repeated children.

2. Compute $\mathbf{g}_1, \cdots, \mathbf{g}_m = \text{SPLIT}_T(\mathbf{h})$. The function $\text{SPLIT}_T(\mathbf{h}) : \mathbb{R}^k \to \mathbb{R}^k \times \cdots \times \mathbb{R}^k$ maps the $k$-dimensional vector $\mathbf{h}$ into $n$ separate $k$-dimensional vectors $\mathbf{g}_1$ to $\mathbf{g}_m$.

3. For each singular child $i$:

    (a) Sample $T_i \sim \text{WHICHCHILDTYPE}_{T,i}(\mathbf{g}_i)$ from a distribution over the types in $\text{REQUIREDCHILD}(T, i)$.

    (b) If $T_i \neq \emptyset$, use $\text{GENERATENODE}(T_i, \mathbf{g}_i)$ to build the child node recursively.

4. If $T$ specifies repeated children:

    (a) Compute $\mathbf{g}_{\text{cur}}, \mathbf{g}_{\text{next}} = \text{SPLITREPEATED}_T(\mathbf{g}_m)$.

    (b) Sample $s \sim \text{STOPREPEAT}_T(\mathbf{g}_{\text{cur}})$ from a Bernoulli distribution. If $s = 1$, stop generating repeated children.

    (c) Sample $T_{\text{child}} \sim \text{WHICHCHILDTYPE}_{T,repeated}(\mathbf{g}_{\text{cur}})$, a probability distribution over the types in $\text{REPEATEDCHILDREN}(T)$.

    (d) Use $\text{GENERATENODE}(T_{\text{child}}, \mathbf{g}_{\text{cur}})$ to build this child recursively.

    (e) Set $\mathbf{g}_m := \mathbf{g}_{\text{next}}$ and repeat this loop.

For building the entire tree starting from the root, we assume that we have an embedding $\mathbf{z}$ which encodes information about the entire tree (we describe how we obtain this in the next section). We sample $T_{\text{root}} \sim \text{WHICHROOTTYPE}(\mathbf{z})$, the type for the root node, and then run $\text{GENERATENODE}(T_{\text{root}}, \mathbf{z})$.

### 3.3 ENCODING A TREE

Recall that the variational autoencoder framework involves training two models simultaneously: $p(\mathbf{x} \mid \mathbf{z})$, the generative (or decoding) model, and $q(\mathbf{z} \mid \mathbf{x})$, the inference (or encoding) model. The previous section described how we specify $p(\mathbf{x} \mid \mathbf{z})$, so we now turn our attention to $q(\mathbf{z} \mid \mathbf{x})$.

Overall, we build the inference model by inverting the flow of data in the generative model. Specifically, we use $\text{ENCODE}(n)$ to encode a node $n$ with type $T$:

- If $T$ is a terminal type, return $\text{EMBEDDING}(v) \in \mathbb{R}^k$ by performing a lookup of the contained value $v$ within a table.

- If $T$ is a non-terminal type:

    1. Compute $g_i = \text{ENCODE}(n_i)$ for each singular child $n_i$ of $n$. If $n_i$ is missing, then $g_i = 0$.

    2. If $T$ specifies repeated children, set $g_{\text{repeated}} := 0$ and $n_{\text{child}}$ to the last repeated child of $n$, and then run:

        (a) Compute $g_{\text{child}} = \text{ENCODE}(n_{\text{child}})$.

        (b) Set $g_{\text{repeated}} := \text{MERGEREPEATED}_T(g_{\text{repeated}}, g_{\text{child}}) \in \mathbb{R}^k$.

        (c) Move $n_{\text{child}}$ to the previous repeated child, and repeat (until we run out of repeated children).

    3. Return $\text{MERGE}_T(g_1, \ldots, g_m, g_{\text{optional}}, g_{\text{repeated}}) \in \mathbb{R}^k$.

Thus $h_{\text{root}} = \text{ENCODE}(n_{\text{root}})$ gives a summary of the entire tree as a $k$-dimensional embedding. We then construct $q(z \mid x) = \mathcal{N}(\mu, \sigma)$ where $\mu = W_\mu h_{\text{root}}$ and $\sigma = \text{softplus}(W_\sigma h_{\text{root}})$. Applying $\text{softplus}(x) = \log(1 + e^x)$ as a nonlinearity gives us a way to ensure that $\sigma$ is positive as required.

### 3.4 IMPLEMENTING **SPLIT**, **MERGE**, AND **WHICH** FUNCTIONS

In the previous two sections, we described how the model traverses the tree bottom-up to produce an encoding of it ($q(\mathbf{z} \mid \mathbf{x})$), and how we can generate a tree top-down from the encoding ($p(\mathbf{x} \mid \mathbf{z})$). In this section, we explicate the details of how we implemented the SPLIT, MERGE, and WHICH functions that we used previously.

**COMBINE.** We can consider $\text{SPLIT} : \mathbb{R}^k \to \mathbb{R}^k \times \cdots \times \mathbb{R}^k$ and $\text{MERGE} : \mathbb{R}^k \times \cdots \times \mathbb{R}^k \to \mathbb{R}^k$ functions to be specializations of a more general function $\text{COMBINE} : \mathbb{R}^k \times \cdots \times \mathbb{R}^k \to \mathbb{R}^k \times \cdots \times \mathbb{R}^k$ which takes $m$ inputs and produces $n$ outputs.

A straightforward implementation of COMBINE is the following:

$$\mathbf{y}_1, \ldots, \mathbf{y}_n := \text{COMBINE}(\mathbf{x}_1, \ldots, \mathbf{x}_m)$$

$$[\mathbf{y}_1 \quad \cdots \quad \mathbf{y}_n] = f(W [\mathbf{x}_1 \quad \cdots \quad \mathbf{x}_m] + \mathbf{b})$$

where we have taken $\mathbf{x}_i$ and $\mathbf{y}_i$ to be column vectors $\mathbb{R}^k$, $[\mathbf{x}_1 \cdots \mathbf{x}_m]$ stacks the vectors $\mathbf{x}_i$ vertically, $W \in \mathbb{R}^{n \cdot k \times m \cdot k}$ and $b \in \mathbb{R}^{n \cdot k}$ are the learned weight matrix and bias vector respectively, and $f$ is a nonlinearity applied elementwise.

For WHICH : $\mathbb{R}^k \to \mathbb{R}^d$, which computes a probability distribution over $d$ choices, we use a specialization of COMBINE with one input and one ($d$-sized rather than $k$-sized) output, and use softmax as the nonlinearity $f$.

While this basic implementation sufficed initially, we discovered that two modifications led to better performance, which we describe subsequently.

**Gating.** We added a form of multiplicative gating, similar to those used in Gated Recurrent Units (Chung et al., 2014), Highway Networks (Srivastava et al., 2015), and Gated Pixel-CNN (van den Oord et al., 2016). The multiplicative gate enables the COMBINE function to more easily pass through information in the inputs to the outputs if that is preferable to transforming the input. Furthermore, the multiplicative interactions used in computing the gates may help the neural network learn more complicated functions.

First, we compute *candidate* values for $\mathbf{y}_i$ using a linear layer as before:

$$[\hat{\mathbf{y}}_1 \quad \cdots \quad \hat{\mathbf{y}}_n] = f(W [\mathbf{x}_1 \quad \cdots \quad \mathbf{x}_m] + \mathbf{b})$$

Then we compute *multiplicative gates* for each $\hat{\mathbf{y}}_i$ and each $(\mathbf{x}_i, \mathbf{y}_j)$ combination, or $(m+1)n$ gate variables (recall that $m$ is the number of inputs and $n$ is the number of outputs).

$$[\mathbf{g}_{y_1} \quad \cdots \quad \mathbf{g}_{y_n}] = \sigma(W_{g_y} [\mathbf{x}_1 \quad \cdots \quad \mathbf{x}_m] + \mathbf{b}_{g_y})$$

$$[\mathbf{g}_{(x_1, y_1)} \quad \cdots \quad \mathbf{g}_{(x_1, y_n)}] = \sigma(W_{g_1} [\mathbf{x}_1 \quad \cdots \quad \mathbf{x}_m] + \mathbf{b}_{g_1})$$

$$\vdots$$

$$[\mathbf{g}_{(x_m, y_1)} \quad \cdots \quad \mathbf{g}_{(x_m, y_n)}] = \sigma(W_{g_m} [\mathbf{x}_1 \quad \cdots \quad \mathbf{x}_m] + \mathbf{b}_{g_m})$$

Then we compute the final outputs $\mathbf{y}_i$:

$$\mathbf{y}_i = \mathbf{g}_{y_i} \odot \hat{\mathbf{y}}_i + \mathbf{g}_{(x_1, y_i)} \odot \mathbf{x}_1 + \cdots + \mathbf{g}_{(x_m, y_i)} \odot \mathbf{x}_m.$$

$\sigma$ is the sigmoid function $\sigma(x) = 1/(1 + e^{-x})$ and $\odot$ is the elementwise product.

We initialized $\mathbf{b}_{g_i} = 1$ and $\mathbf{b}_{g_y} = -1$ so that $\mathbf{g}_{y_i}$ would start out as a small value and $\mathbf{g}_{(x_i, y_j)}$ would be large, encouraging copying of the inputs $\mathbf{x}_i$ to the outputs $\mathbf{y}_i$.

**Layer normalization.** We found that using layer normalization (Ba et al., 2016) also helps stabilize the learning process. For our model, it is difficult to use batch normalization because the connections of each layer (the functions MERGE, SPLIT, WHICH) occur at variable points according to the particular tree we are considering.

Following the procedure in the appendix of Ba et al. (2016), we replace each instance of $f(W [\mathbf{x}_1 \quad \cdots \quad \mathbf{x}_m] + \mathbf{b})$ with $f(LN(W_1 \mathbf{x}_1; \alpha_1) + \cdots + LN(W_m \mathbf{x}_m; \alpha_m) + \mathbf{b})$ where $W_i \in \mathbb{R}^{nk \times k}$ are horizontal slices of $W$ and $\alpha_i \in \mathbb{R}$ are learned multiplicative constants. We use $LN(\mathbf{z}; \alpha) = \alpha \cdot (\mathbf{z} - \mu)/\sigma$ where $\mu \in \mathbb{R}$ is the mean of $\mathbf{z} \in \mathbb{R}^k$ and $\sigma \in \mathbb{R}$ is the standard deviation of $\mathbf{z}$.

## 3.5 WEIGHT SHARING

In the above model, each function with a different name has different weights. For example, if we have two types PLUS and MINUS each with two required children, then SPLIT$_{\text{PLUS}}$ and SPLIT$_{\text{MINUS}}$ will have different weights even though they both have the same signature $\mathbb{R}^k \to \mathbb{R}^k \times \mathbb{R}^k$.

However, this may be troublesome when we have a very large number of types, because in this scheme the amount of weights increases linearly with the number of types. For such cases, we can apply some of the following modifications:

- Replace all instances of $\text{SPLIT}_T : \mathbb{R}^k \to \mathbb{R}^k \times \cdots \times \mathbb{R}^k$ and SPLITREPEATED with a single SPLITREC $: \mathbb{R}^k \to \mathbb{R}^k \times \mathbb{R}^k$. We can apply SPLITREC recurrently to get the desired number of child embeddings.

- Similarly, replace instances of MERGE with MERGEREC.

- Share weights across the WHICH functions: a WHICH function which produces a distribution over $T_1, \ldots, T_n$ contains weights and a bias for each $T_i$. We can share these weights and biases across all WHICH functions where $T_i$ appears.

### 3.6 VARIABLE-SIZED LATENT STATE

In order to achieve low reconstruction error $\mathbb{E}_{q_\phi(\mathbf{z}|\mathbf{x})}[\log p_\theta(\mathbf{x} \mid \mathbf{z})]$, the encoder and decoder networks must learn how to encode all information about a tree in $z$ and then be able to reproduce the tree from this representation. If the tree is large, it becomes a difficult optimization problem to learn how to do this effectively, and may require higher-capacity networks in order to succeed at all which would require more time to train.

Instead, we can encode the tree with a variable number of latent state vectors. For each node $n_i$ in the tree, we specify $q(\mathbf{z}_{n_i} \mid \mathbf{x}) = \mathcal{N}(\mu_{n_i}, \sigma_{n_i})$ where

$$\begin{bmatrix} \mu_{n_i} \\ \sigma_{n_i} \end{bmatrix} = \begin{pmatrix} \text{id} \\ \text{softplus} \end{pmatrix} \begin{bmatrix} W_\mu \\ W_\sigma \end{bmatrix} \text{ENCODE}(n_i)$$

Then when computing GENERATENODE$(T, \mathbf{h})$, we first sample $\mathbf{z}_{n_i} \sim q(\mathbf{z}_{n_i} \mid x)$ at training time or $\mathbf{z}_{n_i} \sim p(\mathbf{z})$ at generation time, and then use $\hat{\mathbf{h}} = \text{MERGELATENT}(\mathbf{h}, \mathbf{z}_{n_i})$ in lieu of $\mathbf{h}$.

We fixed the prior of each latent vector $\mathbf{z}_i$ to be the standard multivariate normal distribution with diagonal unit covariance, and did not investigate computing the prior as a function of other samples of $\mathbf{z}_i$ as in Chung et al. (2015) or Fraccaro et al. (2016) which also used a variable number of latent state vectors.

## 4 EXPERIMENTS

### 4.1 TYPE-AWARE SEQUENTIAL MODEL

For purposes of comparison, we implemented a standard LSTM model for generating each node of the tree sequentially with a depth-first traversal, similar to Vinyals et al. (2015). The model receives each non-terminal type, and terminal value, as a separate token. We begin the sequence with a special ⟨BOS⟩ token. Whenever an optional child is not present, or at the end of a sequence of repeated children, we insert ⟨END⟩. This allows us to unambiguously specify a tree following a given grammar as a sequence of tokens.

At generation time, we keep track of the partially-generated tree in order to only consider those tokens which would be syntactically allowed to appear at that point. We also tried using this constraint at training time: when computing the output probability distribution, only consider the syntactically allowed tokens and leave the unnormalized log probabilities of the others unconstrained. However, we found that for our datasets, this did not help much with performance and led to overfitting.

### 4.2 SYNTHETIC ARITHMETIC DATA

To evaluate the performance of our models in a controlled way, we created a synthetic dataset consisting of arithmetic expressions of a given depth which evaluate to a particular value.

**Grammar.** We have two non-terminal types, PLUS and MINUS, and one terminal type NUMBER. PLUS and MINUS have two required children, *left* and *right*, each of which can be any of PLUS, MINUS, or NUMBER. For NUMBER, we allowed terminal values 0 to 9.

**Generating the data.** We generate trees with a particular *depth*, defined as the maximal distance from the root to any terminal node. As such, we consider $1 + (2 + 3)$ and $(1 + 2) - (3 + 4)$ to both have depth 3. To get trees of depth $d$ which evaluate to $v$, we first sampled 1,000,000 trees

| Depth | Number of nodes | | | Tree, no VAE | Tree VAE | | Tree VAE (var. latent) | | Sequential |
|---|---|---|---|---|---|---|---|---|---|
| | Mean | Min | Max | $\log p(\mathbf{x})$ | $\log p(\mathbf{x}) \geq$ | $\log p(\mathbf{x}) \approx$ | $\log p(\mathbf{x}) \geq$ | $\log p(\mathbf{x}) \approx$ | $\log p(\mathbf{x})$ |
| 5 | 15 | 11 | 19 | -28.26 | -27.03 | -26.85 | -27.02 | -26.86 | **-25.21** |
| 7 | 58 | 39 | 75 | -106.06 | -82.08 | -80.19 | -95.32 | -92.68 | **-74.81** |
| 9 | 206 | 187 | 251 | -332.66 | -331.03 | **-330.68** | -331.12 | -330.78 | -330.75 |
| 11 | 710 | 641 | 1279 | -1172.96 | -1169.85 | **-1169.44** | | | -1404.18 |

Table 1: Statistics of the synthetic arithmetic datasets, and log likelihoods of models trained on them. To estimate a tighter bound for $\log p(\mathbf{x})$, we use IWAE (Burda et al., 2015) with 50 samples of $\mathbf{z}$. "Tree, no VAE" means there was no encoder; instead, we learned a fixed $\mathbf{z}$ for all trees.

| Number of | | Number of nodes | | | Tree, no VAE | Tree VAE | | Sequential |
|---|---|---|---|---|---|---|---|---|
| Functions | Predicates | Mean | Min | Max | $\log p(\mathbf{x})$ | $\log p(\mathbf{x}) \geq$ | $\log p(\mathbf{x}) \approx$ | $\log p(\mathbf{x})$ |
| 6798 | 3140 | 15 | 1 | 2455 | -57.74 | -33.53 | -30.52 | **-29.22** |

Table 2: Statistics for first-order logic proof clauses, and log likelihoods of models trained on them. See Table 1 for more information about the column names.

uniformly at random from all binary tree structures up to depth $d-1$, and randomly assigning each non-terminal node to PLUS or MINUS and setting each terminal node to a random integer between 0 and 9. Then we randomly pick two such trees, which when combined with PLUS or MINUS, evaluate to $v$ to build a tree of depth $d$.

As training data, we generated 100,000 trees of depth 5, 7, 9, and 11. Within each set of trees, each quarter evaluates to $-10$, $-5$, 5, and 10 respectively. We use a test set of 1,024 trees, which we generated by first sampling a new set of 1,000,000 subtrees independently.

**Results.** Table 1 shows statistics on the datasets and the experimental results we obtained from training various models. The tree variational autoencoder model achieves better performance on deeper trees. In particular, the sequential model fails to learn well on depth 11 trees. However, it appears that a tree-structured model but with a fixed $\mathbf{z}$ performs similarly, although consistently worse than with the VAE.

### 4.3 FIRST-ORDER LOGIC PROOF CLAUSES

We next consider a dataset derived from Alemi et al. (2016): fragments of automatically-generated proofs for mathematical theorems stated in first-order logic. An automated theorem prover tries to prove a *hypothesis* given some *premises*, producing a series of steps until we conclude the hypothesis follows from the premises. Many theorem provers work by *resolution*; it negates the hypothesis and shows that a contradiction follows. However, figuring out the intermediate steps in the proof is a highly nontrivial search procedure. If we can use machine learning to generate clauses which are likely to appear as a proof step, we may be able to speed up automated theorem proving significantly.

**Grammar.** We generate first-order logic statements which are *clauses*, or a disjunction of *literals*. Each literal either contains one *predicate* invocation or asserts that two *expressions* are equal. Each predicate invocation contains a name, which also determines a fixed number of arguments; each argument is an expression. An expression is either a *function* invocation, a *number*, or a *variable*. A function invocation is structurally identical to a predicate invocation.

We consider the set of functions and predicates to be closed. Furthermore, given that each function and predicate has a fixed number of arguments, we made each of these its own type in the grammar. To avoid having a very large number of weights as a consequence, we applied the modifications described in Section 3.5.

**Results.** Table 2 describes our results on this dataset. We trained on 955,529 trees and again tested on 1,024 trees. The sequential model demonstrates slightly better log likelihood compared to the tree variational autoencoder model. However, on this dataset we observe a significant improvement

in log likelihood by adding the variational autoencoder to the tree model, unlike on the arithmetic datasets.

## 5 DISCUSSION AND FUTURE WORK

**Conditioning on an outside context.**   In many applications for modeling tree-structured data, we have an outside context that informs which trees are more likely than others. For example, when generating clauses which may appear in a proof, the hypothesis in question greatly influences the content of the clauses. We leave this question to future work.

**Scaling to larger trees.**   Currently, training the model requires processing an entire tree at once, first to encode it into $\mathbf{z}$ and then to decode $\mathbf{z}$ to reproduce the original tree. This can blow up the memory requirements, requiring an undesirably small batch size. For autoregressive sequence models, truncated backpropagation through time provides a workaround as a partial form of the objective function can be computed on arbitrary subsequences. In our case, adapting methods from Gruslys et al. (2016) and others may prove necessary.

**Improving log likelihood.**   In terms of log likelihood, our model performed significantly better than an autoregressive sequential model only on one of the datasets we tested, and about the same or slightly worse on the others. Adding depth to the tree structure (Irsoy & Cardie, 2014), and a more sophisticated posterior (Rezende & Mohamed, 2015; Kingma et al., 2016; Sønderby et al., 2016), are some modifications which might help with learning a more powerful model. Introducing more dependencies between the dimensions of $\mathbf{x}$ during generation is another possibility but one which may reduce the usefulness of of the latent representation (Bowman et al., 2015).

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

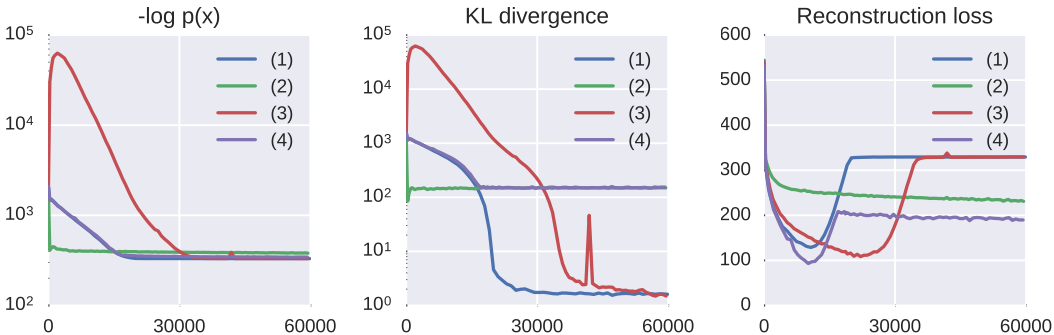

Figure 2: Metrics of the same model trained with different KL-related hyperparameters. The $x$ axis is the step count. (1): We annealed the KL cost weight from step 5000 to 25000. (2): We set the KL cost minimum ($\lambda$) to 150. (3): We annealed the KL cost weight from step 20000 to 40000. (4): We annealed the KL cost weight from 5000 to 25000 and also set the minimum to 150.

## A    MODEL HYPERPARAMETERS

We used the Adam optimizer with a learning rate of 0.01, multiplied by 0.98 every 10000 steps. We clipped gradients to have L2 norm 3. For the synthetic arithmetic data, we used a batch size of 64; for the first-order logic proof clauses, we used a batch size of 256. For the tree-structured variational autoencoder, we set $k = 256$ and used the ELU nonlinearity wherever another one was not explicitly specified. For the sequential models, we used two stacked LSTMs each with hidden state size 256, no dropout. We always unrolled the network to the full length of the sequence during training, and did not perform any bucketing of sequences by length.

## B    KL DIVERGENCE DYNAMICS DURING TRAINING

Optimizing the variational autoencoder objective turned out to be a significant optimization challenge, as pointed out by prior work (Bowman et al., 2015; Sønderby et al., 2016; Kingma et al., 2016). Specifically, it is easy for the KL divergence term $D_{KL}(q_\phi(\mathbf{z} \mid \mathbf{x}) \,\|\, p(\mathbf{z}))$ to collapse to zero, which means that $q_\phi(\mathbf{z} \mid \mathbf{x})$ is equal to the prior and does not convey any information about $\mathbf{x}$. This leads to uninteresting latent representations and reduces the generative model to one that does not use a latent representation at all.

As explained by Kingma et al. (2016), this phenomenon occurs as at the beginning of training it is much easier for the optimization process to move $q_\phi(\mathbf{z} \mid \mathbf{x})$ closer to the prior $p(\mathbf{z})$ than to improve $p(\mathbf{x} \mid \mathbf{z})$, especially when $q_\phi(\mathbf{z} \mid \mathbf{x})$ has not yet learned how to convey any useful information. To combat this, we use a combination of two techniques described in the previous work:

- Anneal the weight on the KL cost term slowly from 0 to 1. Similar to Bowman et al. (2015), our schedule was a shifted and horizontally-scaled sigmoid function.[1]
- Set a floor on the KL cost, i.e. use $-\max(D_{KL}(q_\phi(\mathbf{z} \mid \mathbf{x}) \,\|\, p(\mathbf{z})), \lambda)$ instead of $D_{KL}(q_\phi(\mathbf{z} \mid \mathbf{x}) \,\|\, p(\mathbf{z}))$ in the objective (Kingma et al., 2016). This change means that the model receives no penalty for producing a KL divergence below $\lambda$, and as the other part of the objective (the reconstruction term) benefits from a higher KL divergence, it naturally learns a more informative $q_\phi(\mathbf{z} \mid \mathbf{x})$ at least $\lambda$ in KL divergence.

We found that at least one of these techniques were required to avoid collapse of the KL divergence to 0. However, as shown in Figure 2, we found that different combinations of these techniques could led to different overall results, suggesting that finding the desired equilibrium necessitates a hyperparameter search.

---

[1]To anneal from $a$ to $b$, we used $\sigma\left(\text{step} - \frac{a+b}{2}\right)/10$ to weight the KL cost as a function of the number of optimization steps taken.

