# Peer review of "Tree-Structured Variational Autoencoder"

_ICLR 2017 — rejected_

[Official Review · AnonReviewer4 · rating 3 · confidence 4 · 16 Dec 2016]
**a preliminary look at tree based variational inference**

The authors propose a variational autoencoder for a specific form of tree-generating model.

The generative model for trees seems reasonable but is not fully motivated. If no previous references suggest this tree specification, then clear motivation for e.g. the extension beyond CFG should be given beyond the one sentence provided.

Given the tree model it may be natural to specify a tree model encoder, but the posterior distribution does not respect the structure of the prior (as the posterior distribution couples tree-distant variables), so there is in fact no good reason for this form, and a more general network could be compared with.

The approach provides sensible differentiable functions for encoding the network. The tests are indicative, but the results are very similar to the tested approaches, and it is not clear what the best evaluation metric ought to be.

Significance: the work may well be significant in the future, but is currently somewhat preliminary, lacks motivation, chooses a tree structured encoder without particular motivation, and is lacking in wider comparisons. There is also some lack of current motivation for the model, and no comparison with tractable models that do not need a variational autoencoder.

Originality: original, but at the moment it is not clear such originality is necessary.

Clarity: Good.

Experiments: Sensible, but not extensive or conclusive.

[Official Review · AnonReviewer1 · rating 4 · confidence 4 · 16 Dec 2016]

This paper introduces a novel extension of the variational autoencoder to arbitrary tree-structured outputs. Experiments are conducted on a synthetic arithmetic expression dataset and a first-order logic proof clause dataset in order to evaluate its density modeling performance.

Pros:
+ The paper is clear and well-written.
+ The tree-structure definition is sufficiently complete to capture a wide variety of tree types found in real-world situations.
+ The tree generation and encoding procedure is elegant and well-articulated.
+ The experiments, though limited in scope, are relatively thorough. The use of IWAE to obtain a better estimate of log likelihoods is a particularly nice touch.

Cons:
- The performance gain over a baseline sequential model is marginal.
- The experiments are limited in scope, both in the datasets considered and in the evaluation metrics used to compare the model with other approaches. Specifically: (a) there is only one set of results on a real-world dataset and in that case the proposed model performs worse than the baseline, and (b) there is no evaluation of the learned latent representation with respect to other tasks such as classification.
- The ability of the model to generate trees in time proportional to the depth of the tree is proposed as a benefit of the approach, though this is not empirically validated in the experiments.

The procedures to generate and encode trees are clever in their repeated use of common operations. The weight sharing and gating operations seem important for this model to perform well but it is difficult to assess their utility without an ablation (in Table 1 and 2 these modifications are not evaluated side-by-side). Experiments in another domain (such as modeling source code, or parse trees conditioned on a sentence) would help in demonstrating the utility of this model. Overall the model seems promising and applicable to a variety of data but the lack of breadth in the experiments is a concern.

* Section 3.1: "We distinguish three types" => two
* Section 3.6: The exposition of the variable-sized latent state is slightly confusing because the issue of how many z's to generate is not discussed.
* Section 4.2-4.3: When generating the datasets, did you verify that the test set is disjoint from the training set?
* Table 1: Is there a particular reason why the variable latent results are missing for the depth 11 trees?

[Official Review · AnonReviewer3 · rating 3 · confidence 4 · 17 Dec 2016]
**No Title**

The method overall seems to be a very interesting structural approach to variational autoencoders, however it seems to lack motivation as well as the application areas sufficient to prove its effectiveness.

I see the attractiveness of using structural information in this context and I find it more intuitive than using a flat sequence representation, especially when there is a clear structure in the data. However experimental results seem to fail to be convincing in that regard.

One issue is the lack of a variety of applications in general, the experiments seem to be very limited in that regard, considering that the paper itself speaks about natural language applications. It would be interesting to use the latent representations learned with the model for some other end task and see how much it impacts the success of that end task compared to various baselines.

In my opinion, the paper has a potentially strong idea however in needs stronger results (and possibly in a wider variety of applications) as a proof of concept.

[Final Decision · Program Chairs · 06 Feb 2017]
**ICLR committee final decision**

This paper is clearly written, but the method isn't demonstrated to solve any problem much better than simpler approaches. To quote one reviewer, "the work may well be significant in the future, but is currently somewhat preliminary, lacks motivation, chooses a tree structured encoder without particular motivation, and is lacking in wider comparisons."